# The Development of Technogenic Deposits as a Factor of Overcoming Resource Limitations and Ensuring Sustainability (Case of Erdenet Mining Corporation SOE in Mongolia)

Ivan Potravny [1], Andrey Novoselov [2], Irina Novoselova [3], Violetta Gassiy [4,*] and Davaakhuu Nyamdorj [5]

1 Basic Department of Project and Program Management Capital Group, Plekhanov Russian University of Economics, 36 Stremyanny Lane, 117997 Moscow, Russia; ecoaudit@bk.ru
2 Department of Economics of Oil and Gas Industry, Gubkin University, 65 Leninsky Prospekt, 119991 Moscow, Russia; alnov2004@yandex.ru
3 Department of Industry Markets, Financial University under the Government of the Russian Federation, 49/2 Leningradsky Prosp., 125167 Moscow, Russia; iunov2010@yandex.ru
4 Public Administration Department, Kuban State University, 149, Stavroposkaya, 350040 Krasnodar, Russia
5 Science Department, Ulaanbaatar Branch of the Plekhanov Russian University of Economics, Bayanzurkh District, 14th Section, Mira Avenue, No. 69, Ulaanbaatar 13335, Mongolia; dabuk91@mail.ru
* Correspondence: vgassiy@mail.ru

**Abstract:** This article justifies the need to involve technogenic deposits (off-balance ore and wastes) into the economic circulation of mining enterprises when there is a depletion of natural resources. It could be considered as one of the tools of the circular economy. The authors analyze global trends in the development of copper deposits, global demand for copper, and design recommendations for possible alternative options for the copper production. The authors use the case of Erdenet Mining Corporation SOE based in Mongolia to develop the approach for economic, social, and environmental problem-solving. The millions of mining dumps are proposed to develop as technogenic resources for recycled materials, prolonging profitable activities of the mine. The hierarchy analysis method is used to obtain the optimum order of mining dump development to obtain the desired economic, social, and environmental effect.

**Keywords:** depletion of natural capital; mining; technogenic deposits; mining dumps; circular economy; environmental protection; Erdenet Mining Corporation SOE; Mongolia

## 1. Introduction

Natural capital depletion and waste accumulation are among the main global environmental problems, as well as climate change; air, water, and soil pollution; and loss of biodiversity [1]. Preservation of natural capital and ensuring green economic growth are becoming the main task in the context of the depletion of the production resource base. It is possible to solve by optimizing nature management through the inclusion of production waste and the resources of technogenic deposits into economic circulation [2]. These changes related to the exhaustion of raw materials and the generation of waste are limiting factors for sustainable development. Therefore, the theory and practice of modern economic development pays great attention to the principles of resource conservation. A promising model of environmental management is based on the use of waste to preserve natural capital, to reduce production costs, and to improve the environment and living conditions of the population.

Nowadays, there are global challenges to environmental security such as the increasing consumption of natural resources and the reduction in their reserves. This leads to a struggle for the limited natural resources and has a negative impact on the economy, population, and the environment. Additionally, the resource-based model of economic development has been exhausted and inefficient. Under these conditions, the priority

tasks are processing production waste, increasing the mineral resource base for mining by including secondary material resources into the economic circulation, and reducing the burden on the environment. This approach is especially relevant for the mining sphere. At present, there is already some experience in using technogenic deposits as a new resource base, for example, mine waste rocks formed during diamond mining to extract gold and molybdenum [3].

Modern science has many publications on the problems of accumulated environmental damage and the issues of the depletion of natural capital; therefore, the search for alternative sources of production has begun. Technogenic resources, i.e., those obtained as a result of industrial activity, represent not only the potential for the production of "indirect" products of industrial production or subsoil use, but also secondary ones, for example, metals of smaller fractions can be extracted from dumps during their processing or resources can be obtained for construction (blocks, mixtures for filling roads). An important area of such research is the issue of environmental safety in the future. K. Lapakko considers industrial waste dumps as a potential source of environmental pollution, since water drainage leads to mixing with groundwater and the threat of chemicals entering it increases significantly. The solutions on how to predict the composition of water in dumps in the future in order to reduce the threat of environmental pollution are offered in [4].

Collins, R. J. and R. H. Miller are exploring the possibility of recycling industrial waste dumps, recognizing that there are significant limitations to their use in the production of various products. For example, the elements contained in dumps may be finely dispersed and oxidize, or the dump itself may be too far away and transporting processed waste may not make economic sense. However, such elements can be used in the construction of railway tracks and concreting platforms, including helicopter platforms in remote areas. Each case must be analyzed taking into account such criteria [5].

Smith L. explores the management of mining waste, dividing it into main types. Depending on the species, the author proposes processing mechanisms, assessing their behavior in the future and the impact on the environment and their economic potential. The main challenge for Smith L. is trying to predict the effects of waste processing and the time period [6]. N. Chaturvedi and colleagues associate the problem of waste processing with the costs and inefficiency of available technologies. They believe that the mining of metals and other ores ensures the social well-being of the population, as it serves for the production of goods. However, the downside of this process is the accumulation of huge amounts of waste. According to experts' calculations, more than 700 kg of metals were dumped around the world in 1995. Considering that since then, the volume of subsoil use has increased, the accumulation of useful elements in industrial waste has grown exponentially. Therefore, to increase the economic efficiency of waste disposal, investment in innovation, waste management, and recycling technologies is necessary. The profitability of the mining enterprise as well as social (maintain or even increase employment and income) and environmental (reduce the source of pollution) issues could be solved. N. Chaturvedi discusses the disturbed lands and their restoring based on the case of iron ore deposits in India where phytoremediation methods are used [7].

B. Vriens, B. Plante, N. Seigneur, and H. Jamieson note that rock waste can negatively affect the environment [8]. Xie, M., Liu, F., and Zhao, H., using the case of coal development projects, considered the methods of environmental protection as well as mechanisms of clean and efficient utilization of coal [9]. Techniques for the hydrogeochemical behavior of mine waste rock are reviewed for sustainable management of tailings ponds to optimize management and minimize potential negative environmental impacts [10]. The environmental pollution problem-solving is researched by R. Suppes, S. Heuss-Aßbichler [11]. They analyze the tools for high sulfur bauxite and its phase transformation during desulfurization. The results of the research are based on China's experience but could be paid attention to worldwide.

A significant scientific and practical interest is presented in the case of extracting and developing the tailing dumps in the Republic of Sakha (Yakutia) [11,12]. There is

a huge waste dump of a gold recovery plant at the Kular deposit in the Russian Arctic. The problem of accumulated environmental damage is solved as well [13]. This makes it possible to reduce the risks of environmental emergencies in the Arctic region to obtain useful products, and also to harmonize the interests of the mining company and the local population [14]. In addition, this approach has been improved and a mechanism for interaction and cooperation between target groups, i.e., businesses, local governments, and indigenous peoples during the industrial development of the Arctic, was developed [15].

Currently, in world practice, significant experience has been accumulated in the use of geochemical technologies for processing waste from mining enterprises and obtaining useful products [16].

Modern studies show that the technogenic raw materials involvement into processing can significantly strengthen the economic potential and provide a solution to many urgent problems of subsoil use, including the following:

- more complete use of non-renewable natural resources;
- the preservation of depleting mineral raw materials in the subsoil;
- increase in labor productivity through cost-effective processing of extracted raw materials;
- ensure there is employment by making new jobs;
- cheap building materials (sand, crushed stone, gravel);
- mineral additives to improve soil structure and fertilizers for agriculture;
- reduction or elimination of environmental pollution sources;
- recultivation of lands occupied by waste [17].

One of the factors affecting the functioning of mining enterprises is the possible depletion of mineral reserves, which leads to the closure of the mine and the liquidation of the enterprise. At the same time, the accumulated waste due to mine activities can be considered as a resource base, a kind of technogenic deposits that could be involved to the economic circulation, e.g., for the building materials production. The mine waste rocks formed during the production of marketable ores contain a significant number of useful elements. The authors propose to use the substitute resources of mining, which is understood as a set of technogenic resources (off-balance ore, mine waste rocks, enrichment waste) suitable for reuse as a new raw material base. Such an approach enables the extension of the life of a mining enterprise, reduces the burden on the environment, and obtains the demanded products through recycling. For example, 1 ton of non-ferrous metals accounts for up to 100–150 tons of waste and another 50–60 tons for processing.

Mining waste is a unique source of many valuable rare metals. The exploitation of technogenic deposits makes it possible to maintain the required level of metal production even with a significant decrease in the production of metal ores. It should be noted that industrial wastes are often close in composition and properties to natural raw materials, and in some cases, they have an advantage [18]; one such example is copper.

The depletion of copper is currently due to the fact that many mining enterprises are working on enrichment technologies designed for high and medium grades of minerals in ores. In the near future, according to forecasts, the copper content in ores will decrease to 0.5% and below. At the same time, the mine waste rocks contain large reserves of substandard, poor, and oxidized ores. They are independent technogenic deposits. Calculations made for the Russian Kalmakyr copper site of mine waste rocks leaching showed the economic feasibility of introducing this technology. The need to develop and to implement the projects and programs for waste processing is very up-to-date because in Russia, Mongolia and worldwide, countries are faced with the problem of resource base depletion. In addition, there is an acute issue of eliminating the accumulated damage and the environmental protection.

An analysis of world experience shows that the extraction of minerals and metals has a negative impact on the environment [19]. At the same time, the accumulated waste of mining enterprises can be considered as a kind of technogenic resources that are quite cost-effective to include in economic circulation. This meets the principles of a green and circular economy.

The purpose of this study is to substantiate the approach for the inclusion of techno-genic deposits as a new resource base of a copper mining enterprise, which allows its life to be extended, the processing of useful products through waste management and circular economy development, the reduction of environmental pollution, ensures sustainable development, and keeps local jobs. The proposed approach is justified upon the case of the Erdenet mining enterprise in Mongolia.

## 2. Materials and Methods

### 2.1. The Substitute Resources' Analysis in Mining

The model of mining waste involvement in economic circulation can be built on the basis of replacing primary natural raw materials with substitute resources, which can be, for example, technogenic deposits formed as a result of mining activities. Evaluation of their use helps to determine the relative savings and the benefits of the finished products' production from technogenic resources. In practice, the reduction in available reserves of natural raw materials and resource base depletion in mining is a general trend in the world. This circumstance is already reflected in the financial indicators of such enterprises, including growth of costs for the environment recovery. The development of the industry for the extraction and production of non-ferrous metals, including copper, is based on their involvement in economic circulation and subsoil use of the resources, which are also limited and exhaustible. In many countries, the main problem for mining enterprises' supply of raw materials for copper production is associated with the depletion of mineral reserves, which led to the closure of mines and a reduction in copper production (Table 1).

**Table 1.** The production reduction and closure of mines for the extraction of copper ore due to the depletion of reserves in the world [20].

| Country | Mines | Capacity, Thousand Tons | State of the Mine |
|---|---|---|---|
| USA | Miami, Freeport | 26 | Closed |
| USA | Tyrone, Freeport | 43 | Reduced up to 50% |
| Chile | El Abra, Freeport | 166 | Reduced up to 50% |
| Chile | Collahuasi, Glen/Anglo American | 445.25 | Capacity reduction to 30 thousand tons |
| USA | Ray, Asarco | 53.29 | Capacity reduction to 40% |
| Congo | Katanga, Glencore | 160 | Temporary closed |
| Zambia | Morani, Glencore | 110.185 | Temporary closed |
| Botswana | Mowana | 10 | Closed |

In some countries, such as Chile, Australia, China, Canada, USA, and New Zealand, there is an experience on copper manufacturing from production waste [21]. In terms of explored copper deposits, Russia accounts for 5% of world reserves and Mongolia accounts for 3%. For comparison: less than a third of technogenic waste is recycled in Russia, while in the world, this figure reaches 85–90%. Therefore, an important issue of environmental economics is the analysis and trend assessment of sustainable development of the copper industry in the context of the depletion of raw materials.

Thus, a common trend for Russia and Mongolia in copper production is the reduction of raw materials reserves against the backdrop of an increase in copper demand for national economies. Another trend is the cessation of mining enterprises due to the closure of mines. Therefore, for mining companies in Russia, Mongolia, and other countries, there is a strong search for new resources of the production by using secondary raw materials, as well as problem-solving of the accumulated environmental damage. For example, as a result of the past activities of many mining enterprises in Russia, huge numbers of the accumulated

environmental damage and disturbed territories were formed. They negatively impact the environmental living conditions of the population living in such areas.

This is why the case of Erdenet Mining Corporation SOE is interesting for theory and practice, since in Mongolia, the mining sector in the medium term will remain a key factor in the country's economic growth. Erdenet Mining Corporation SOE is one of the largest enterprises in Asia for the extraction and dressing of copper and molybdenum. It was founded in 1978 at the copper-molybdenum deposit called "Erdenetiin Ovoo". The enterprise provides socio-economic development of the cities of Erdenet (90,000 population) and Darkhan (76,000 population). Geological exploration of the Erdenetiin Ovoo deposit began in the late 1950s. In the period from 1964 to 1968, this field was discovered. In 1976, the first explosion was carried out and an overburden had started; more than 18 million m$^3$ of rock mass per year is mined at the Erdenetiin Ovoo deposit by the open pit method. Currently, this enterprise produces 530.0 thousand tons of copper and more than 4.0 thousand tons of molybdenum concentrate. This enterprise processes 25 million tons of ore per year, and produces 530.0 thousand tons of copper and 4.5 thousand tons of molybdenum concentrate.

The Table 2 shows that Mongolia's mining sector makes a significant contribution to the country's GDP, accounting for a quarter of its income.

**Table 2.** Indicators of heavy industry development in Mongolia in 2021 [22].

| Indicator | Share |
|---|---|
| Mining sector in gross domestic product (GDP) | 24% |
| Industrial sector in national economy | 69% |
| Foreign direct investment | 77% |
| Export | 93% |

Currently, the Erdenetiin Ovoo deposit is based on reserves of porphyry copper ores of the north-western section with a copper content of 0.54–0.57%, and a molybdenum content of 0.0165–0.0179%. At the same time, the waste rock mass contains significant residual amounts of copper, molybdenum, rhenium, silver, and other associated metals, as well as flotation reagents used in technological processes. According to experts, at this mining enterprise, the explored reserves of copper and molybdenum remain for approximately 25–30 years using the current mining technology. The negative consequences are linked to useful components decreasing in raw materials' content, growing the costs of products. At the same time, there is a growing demand in the world, and there is a trend towards an increase in copper production.

Stabilization of world prices for raw materials, primarily for copper, to a large extent stimulated the implementation of waste processing projects and the involvement of off-balance ore in the economic turnover, which, given the low price of primary natural resources, was not profitable enough. This makes it possible to cost-effectively involve off-balance ores and the resources of technogenic deposits into economic circulation. Among the tools for reducing the environmental and economic costs of production and the processing cost, raw materials could be replaced by other, less-expensive resources.

The following case on sustainable mining could be considered. The data for the article and math calculations were the materials of Erdenet Mining Corporation SOE. To obtain finished products (copper and molybdenum) at the same level when the content of copper and molybdenum in the ore is reducing to 0.54% and 0.0167%, a further increase in ore production is necessary. At the same time, the cost of processing mined ore and producing finished products increases every year. The completed SWOT analysis showed that the strengths of Erdenet Mining Corporation SOE include the availability of substitute resources for the production and technogenic deposits as a new resource base for production. The weaknesses of the enterprise are raw material depletion, the need to implement compensatory measures due to mine's closure, the dependence on world prices for raw materials, the lack of facilities for processing copper concentrate, and rising environmental

costs. Among the threats are the closure of the enterprise's mine in the future, instability of prices on the world market, the rising costs of employment, and environmental modernization of production. From the moment the enterprise was created and reached its design capacity, the copper content in ore indicators ranged from 0.685% to 0.833%. Later, it steadily declined from 2013 to 0.530%, and currently stands at 0.45%. Experts project it will drop to 0.21% by 2040. At the same time, in order to maintain copper concentrate production volumes, the company has started increasingly mining and processing ores with worse mineral content. Thus, when working in the combined north-western and central sections with a processing capacity of 35 million tons per year, the life of the mine can be (Figure 1):

(a)     with a cut-off copper content of 0.25%—36 years,
(b)     with a cut-off copper content of 0.15%—41 years.

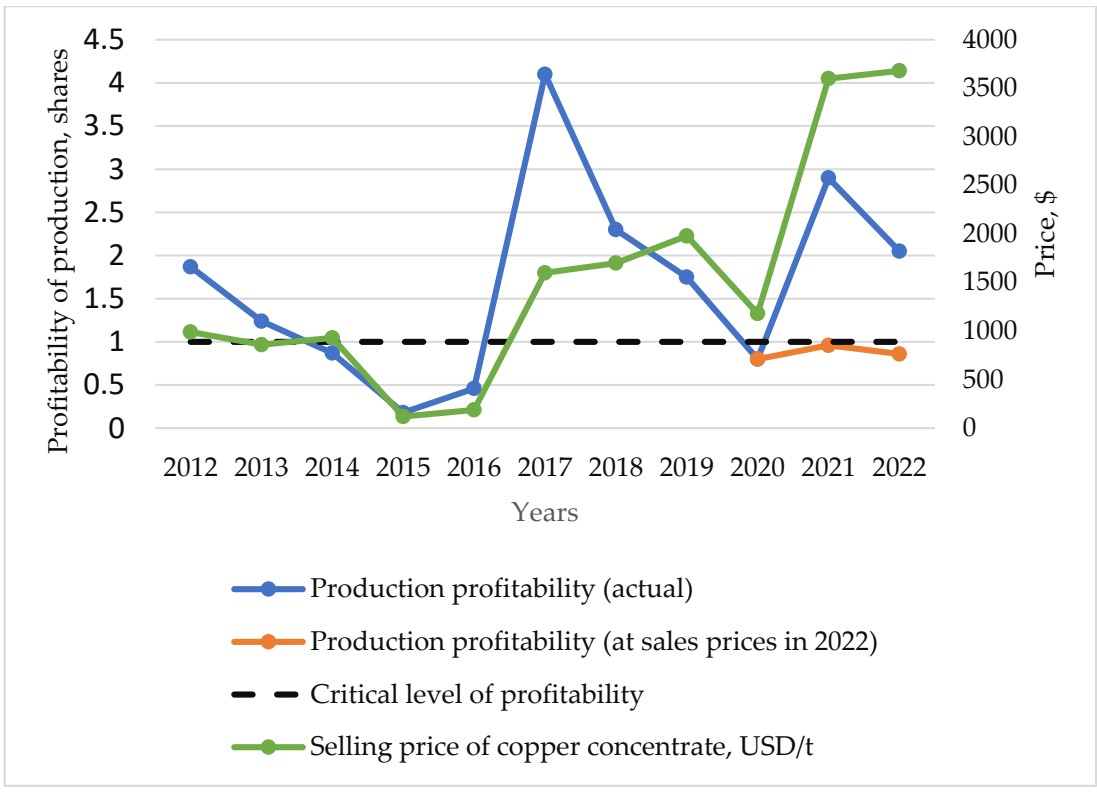

**Figure 1.** Changes in production profitability and selling prices of copper concentrate.

It is obvious that in order to maintain production volumes of finished products in conditions of a copper content decrease in the extracted raw materials, the extraction of a significantly larger volume of rock mass and ore will be required. It will be accompanied by an increase in the cost of its processing. At the same time, there is an increase in volumetric indicators associated with the extraction and processing of ore, while simultaneously reducing the ore content. At the beginning of 2018, the price of 1 ton of copper on world commodity exchanges exceeded USD 7000 (the selling price of copper concentrate from Erdenet Mining Corporation SOE was USD 1730/t). In 2017, rock production at the enterprise increased by 1810 thousand m3 relative to 2010, but ore processing remained at the same level. The production of finished products (copper) decreased significantly.

S&P Global forecast copper growth will rise to 50 million metric tons per year by 2035, exceeding total global consumption from 1990 to 2021. According to estimates by the Canadian mining corporation Ivanhoe Mines, an era of global copper shortage is coming and, in the face of growing unmet demand for this resource, prices will increase by an order of magnitude. The global trend in demand and prices for refined copper shows continuous

growth against the backdrop of a significant depletion of deposits and a reduction in balance reserves of copper ores. Mineralized overburden rocks, substandard off-balance ores, and difficult-to-process oxidized and mixed ores are deposited in the mining waste rocks [21]. Despite the difficulties of extracting copper from deposit dumps, this process is possible and economically feasible with the present technological development and taking into account the constant increase in copper prices.

The exploitation period depends on the size of the deposit and the ore content. At the same time, during the extraction process, the percentage of the ore content decreases. On average, the copper content in ores at developed deposits, for example, in Russia, is reduced to 0.5%. For example, the Gumeshevsky copper mining deposit in Russia has been in operation for 315 years; during this period, copper extraction technologies have changed many times, which has kept mining profitable.

Dumps of copper mining rocks are significant technogenic deposits, consisting of substandard, poor, and oxidized ores' reserves [23]. Existing leaching technologies help to use dumps as technogenic deposits and cost-effective copper extraction areas. The life of the field could be extended, as seen in Figure 2.

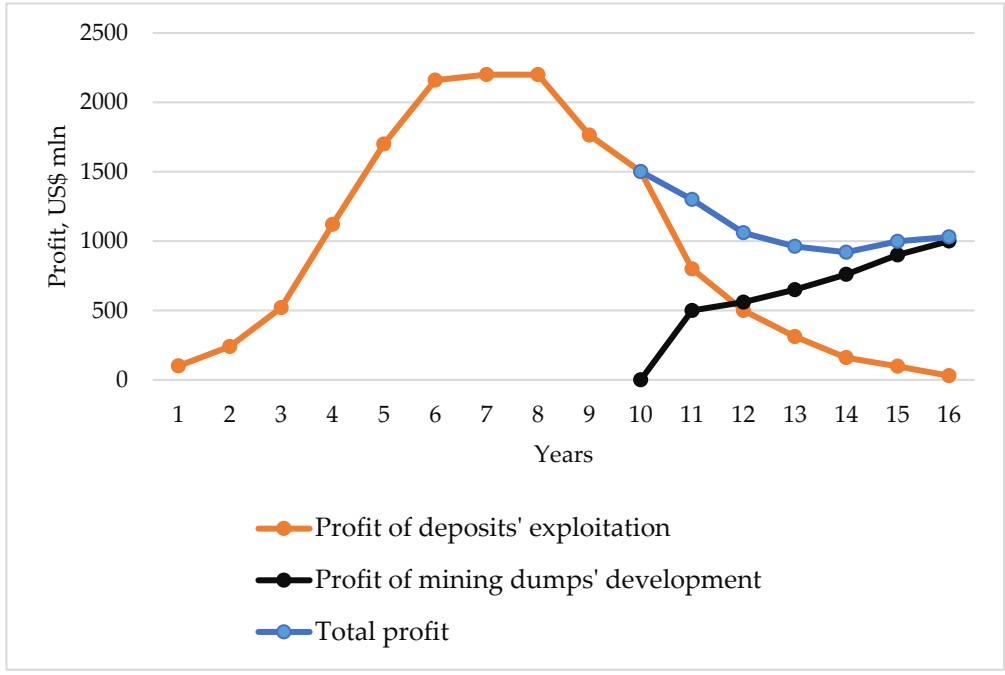

**Figure 2.** Schematic diagram of the economic feasibility of deposit dumps involvement in the process of copper ore mining.

The most technologically simple and least expensive methods are dump or heap leaching, which are successfully implemented in many countries. An enlarged technical and economic calculation made for the copper dump leaching of the Kalmakyr deposit has showed the economic feasibility of introducing this technology. Figure 3 shows the dynamics of copper content in the ore of the Erdenet field. It has decreased from 0.83% to 0.35% during the period of subsoil use. The percentage of copper in the ore dumps is 0.38%, and in the oxidized dumps is 0.53%.

In 2012, the percentage of copper in the ore of the deposit and the ore of the dumps was equal. In 2021, the copper content in the ore of the deposit was at the level of copper content in the oxidized dumps. It is estimated that the average period for technogenic deposit development is 25–30 years. The main task of copper ore deposit dumps' involvement is to rationally determine the procedure for mining based on economic, social, and environmental goals.

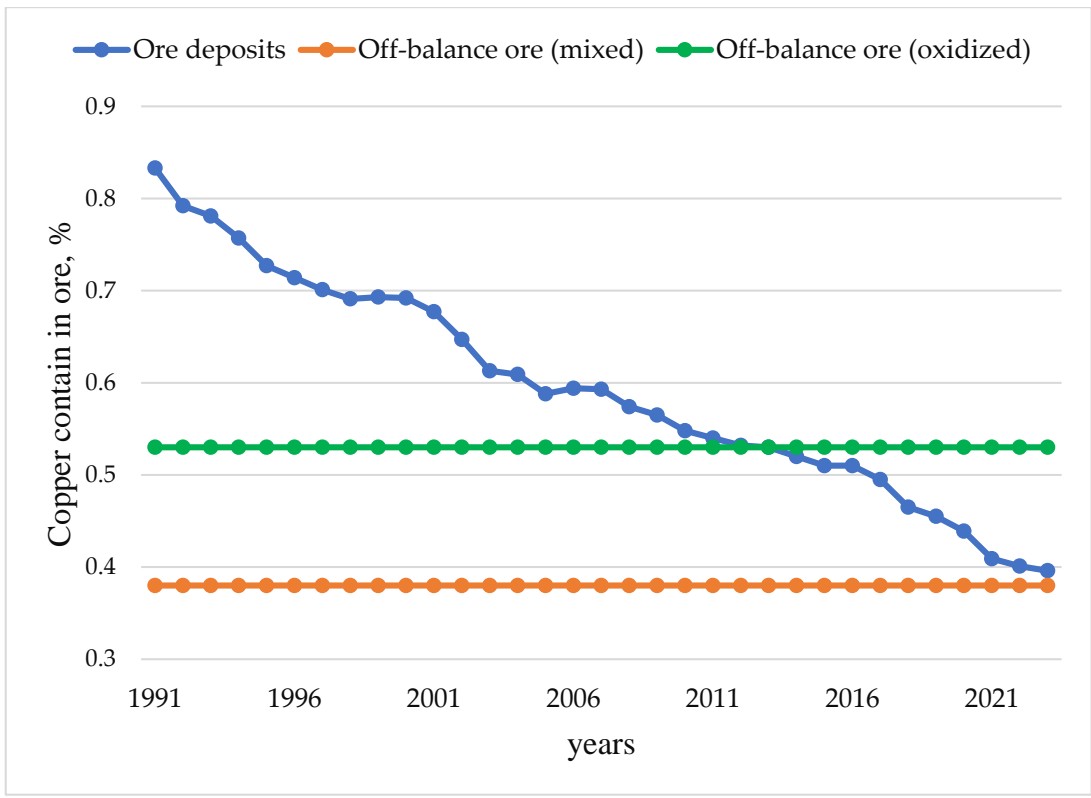

**Figure 3.** Dynamics of copper content in ore during the exploitation of the Erdenetiin Ovoo deposit, Mongolia.

### 2.2. Algorithm of Priority and Timing Assessment for Dumps' Use

The problem-solving requires the development of evaluation criteria each of the goals. The composition of the criteria may be different and should be determined based on the individual characteristics of the enterprise, including remoteness from populated territories occupied by dumps and tailings areas, and their impact on the environment. The following criteria were identified during a discussion with Erdenet Mining Corporation SOE and civil activists living in the mining area:

- Economic: profitability and volume of copper concentrate extraction;
- Social: the demand for the territory occupied by mining dumps and its perspectives for local development;
- Environmental: negative impact on the environment and possible risk of an emergency.

The specifics of the Erdenet production site require annual capital expenditures to maintain and develop. The profitability of the production is determined as a ratio of annual profit to capital costs. The economic goals' assessment criteria are calculated as profitability in shares and the volume of concentrate recovery—in thousand tons during the period of dump development. The social and environmental goals' criteria are assessed by experts using the Saaty scale [23]. A set and number of criteria used for dumps evaluation does not affect the proposed process for problem-solving.

To determine a rational procedure for mining dumps of a copper ore deposit, a calculation procedure has been developed. It is performed by two stages. At the first stage, it is necessary to determine the priority of using dumps, based on all applicable criteria. During such an assessment, all criteria must be used simultaneously with their weight. Among well-known ranking methods, the analytic hierarchy process (AHP) satisfies the formulated conditions. To apply this method, you should use the Saaty scale, which is already included in the expert assessment of socio-ecological criteria [24,25].

Therefore, it is necessary to convert the evaluation criteria into this scale [26–36], to quantify an economic goal using an incremental step based on formula:

$$\Delta = 0.125 \times \left[ \max_{i=1,2,\dots m} (f_i) - \min_{i=1,2,\dots m} (f_i) \right] \tag{1}$$

where $f_i$—criterion value for the $i$-th dump $(i = 1, 2, \dots m)$.

Using the found step $\Delta$, an individual scale ranging from the minimum to maximum value for the used criterion can be obtained. It corresponds to the Saaty scale. For example, if the maximum value of production profitability among the analyzed dumps is 4.22, and the minimum is 1.36, then the step value is $\Delta = 0.125 \times [4.22 - 1.36] = 0.36$. Based on this step, a gradation of production profitability has been obtained. It is associated with lexical units and the Saaty scale.

The algorithm for the priority determining of compared objects based on the hierarchy analysis method is known and widely used in practical calculations [30–35]. It is not advisable to dwell on it. The use of the hierarchy analysis method makes it possible to obtain priority $\rho_i$ for the dumps' development $i = 1, 2, \dots m$.

The second stage of problem-solving is to determine the rational sequence of dump mining [22]. It will ensure the first-priority selection of the highest priority dumps based on the annual volume of allocated financial resources $B_t$ in year $t$ to ensure annual capital costs $Z_i$ for mining maintenance and development during dumps' processing $i = 1, 2, \dots m$. The sequence of dump mining is identified by the required start $T_i^s$ and end $T_i^f$ dates for dump mining with a duration of $t_i$ $i = 1, 2, \dots m$. The heuristic algorithm is developed and used for the rational sequence of dumps' mining determination:

Step 1. Specifying a set of analyzed dumps for which rational sequence and timing of mining $J = \{1, 2, \dots m\}$ has to be found.

Step 2. Setting the current time $\tau = 1$ and values for the volume of financing of capital investments $B_\tau$, $\tau = 1, 2, \dots \max_{i=1,2,\dots m} \left\{ T_i^f \right\}$

Step 3. Selection of the dump $i^*$ with the highest priority within the available financial resources to ensure capital costs for the mining maintenance and development:

$$i^* = ind \left\{ \max_{i \in J} (\rho_i) \wedge Z_i \leq B_\tau \right\} \tag{2}$$

$ind$—operation of selecting the index (dump number) at which the maximum value is achieved $\rho_i$ if financing is possible at the current time $\tau$, i.e., $Z_i \leq B_\tau$.

Step 4. Check: if such a dump is not selected, then move on to the next year $\tau = \tau + 1$; go to step 3. Otherwise, go to step 5.

Step 5. Calculation of time for dump development $i^*$:

$$T_{i^*}^s = \tau \text{ и } T_{i^*}^f = T_{i^*}^s + t_{i^*} - 1 \tag{3}$$

Step 6. Adjustment of remaining funding amounts after choosing a dump $i^*$:

$$B_t = B_t - Z_{i^*} \text{ for } t = \tau, (\tau + 1), \dots (\tau + t_{i^*} - 1) \tag{4}$$

Step 7. Elimination $i^*$ from the set of unselected dumps $J = J - i^*$

Step 8. Check whether all dumps are included in the mining sequence, i.e., $J = \varnothing$? If yes, then the calculations are complete; otherwise, go to step 3. The proposed algorithm can be easily implemented as a *VBA-Excel* macro.

## 3. Results

Based on the proposed scientific tools, the possibility and necessity of technogenic deposits involvement as a new resource base for copper production is substantiated at the Erdenet mining enterprise.

The Erdenet mine dumps located at a distance of 5 to 8 km from urban areas have been chosen as the objects of the research. The plan of their processing must consider economic, social, and environmental priorities. There are two kinds of dumps:

-       with average ore content about 0.38% (dumps of off-balance ores no. 2, 4a, 8, 9);
-       with average ore content 0.53% (dumps of oxidized off-balance ores no. 8a, 12, 2b).

The economic indicators of the analyzed dumps are given in Table 3.

**Table 3.** Economic indicators of the analyzed dumps of the Erdenet deposit.

| Analyzed Dumps | Potential Copper Reserves in Dumps, Thousand Tons | Duration of Dump Development, Years | Capital Investments for Mining Maintenance and Development, Million USD/Year | Production Profitability, Shares |
|---|---|---|---|---|
| Dump N2 | 210 | 7 | 36 | 3.02 |
| Dump N4a | 10 | 2 | 12 | 1.36 |
| Dump N8 | 80 | 4 | 17 | 4.22 |
| Dump N9 | 120 | 6 | 18 | 3.99 |
| Dump N8a | 50 | 5 | 25 | 1.39 |
| Dump N12 | 10 | 2 | 10 | 1.63 |
| Dump N2b | 5 | 1 | 11 | 1.48 |

Based on the Erdenet enterprise data, an assessment of the environmental and social risk of technogenic deposits development to obtain useful products (copper) was carried out. The expert assessment of analyzed dumps according to environmental impact criteria must be also considered. The demand for the territory for local development and the risk of an emergency is given in Table 4.

**Table 4.** Social and environmental assessments of the liquidation of the analyzed dump areas.

| Analyzed Dumps | Assessment of Socio-Ecological Criteria in the Lexical Scale | | |
|---|---|---|---|
| | Impact on the Environment | Demand for the Territory for City Development | Risk of an Emergency |
| Dump N2 | Very low | Average | Average |
| Dump N4a | Very low | High | Average |
| Dump N8 | Low | High | Short |
| Dump N9 | Very low | Very high | Average |
| Dump N8a | Low | Average | Short |
| Dump N12 | Very low | High | Short |
| Dump N2b | Low | Average | Average |

At stage 1, the priorities for the use of mining dumps are assessed by the hierarchy analysis method. Table 5 shows the weights of the criteria (last row of the table) and mountain dumps' assessments for each of the criteria separately (columns 2–6 of the table). In the last column, the priority of mountain dumps' processing is calculated based on weights of the criteria. Taking into account the proposed environmental, social, and other criteria, the priority for the technogenic deposits development at the Erdenet enterprise is proposed.

**Table 5.** Resulting assessments of the priority of mountain dumps' use as technogenic raw materials.

| Analyzed Dumps | Criteria Scores | | | | | Dumps' Priorities |
|---|---|---|---|---|---|---|
| | Profitability of Production | Potential Reserves of Copper | Demand for the Territory. Occupied by Dumps | Risk of an Emergency | Environmental Pollution | |
| Dump N2 | 0.15 | 0.45 | 0.05 | 0.20 | 0.06 | 0.15 |
| Dump N4a | 0.04 | 0.03 | 0.17 | 0.21 | 0.05 | 0.13 |
| Dump N8 | 0.36 | 0.19 | 0.17 | 0.05 | 0.28 | 0.18 |
| Dump N9 | 0.34 | 0.19 | 0.33 | 0.22 | 0.05 | 0.19 |
| Dump N8a | 0.04 | 0.07 | 0.05 | 0.05 | 0.23 | 0.10 |
| Dump N12 | 0.04 | 0.04 | 0.17 | 0.05 | 0.06 | 0.07 |
| Dump N2b | 0.04 | 0.03 | 0.05 | 0.22 | 0.27 | 0.18 |
| Criterion weight | 0.10 | 0.06 | 0.15 | 0.39 | 0.30 | 1.00 |

The last column of Table 5 shows the points, the sum of which is equal to 1. Moreover, dumps number 9, 8, and 2b have the highest priority (0.19, 0.18, and 0.18 points, respectively).

At stage 2, based on the received priorities, capital costs, duration of mining, and the amount of funding, the priority of mining these dumps is determined. Erdenet Mining Corporation SOE has a maximum annual financing of capital costs as USD 62 million for the dumps' development. The year 2024 was adopted as the year for the start of the dumps' mining. For 2024, the highest priority options are selected based on developed algorithm for priority of mining dumps' determination. Dump N9 is selected because it has the highest priority (0.19). Since then, the mining dump's development requires 6 years (Table 3). Figure 4 marks the period from 2024 to 2029 with available financing USD 44 million (because $B_t = 62 - 18 = 44$ for $t = 2024, 2024, …2029$). A forecast has been made for the use of man-made deposit resources at the Erdenet enterprise by replacing primary natural resources (mined ore) for the period until 2036.

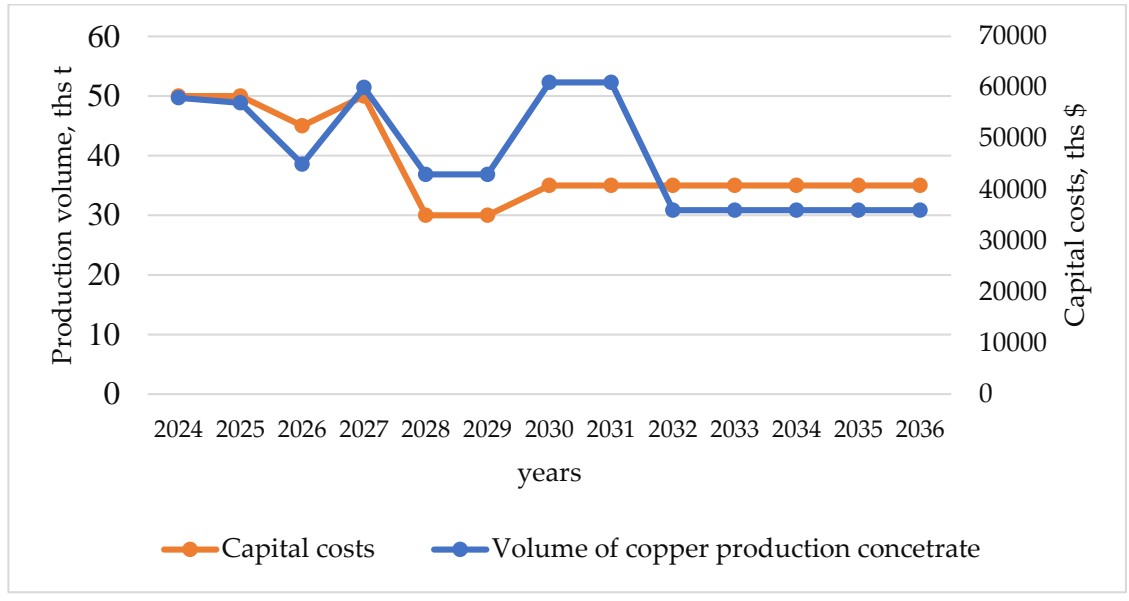

**Figure 4.** Dynamics of economic indicators of dumps mining development in accordance with the obtained order.

For 2024, the N8 mining dump is selected as the highest priority. The capital investment costs are USD 17 million. Since the remaining funding allows for the N8 dump

development in 2024, the process of its mining could be illustrated on a graph. Figure 5 shows the process of mining the N8 dump for the period of 2024–2027. The funding remains have to be adjusted: $B_t = 44 - 17 = 27$ for $t = 2024, 2024, ...2027$. Next, the mining dump N2b is selected with a priority of 0.18. A total of USD 11 million of the capital investments are required for the dump's development. The selected dump requires one year of implementation and is included in the 2024 plan $B_{2024} = 27 - 11 = 16$.

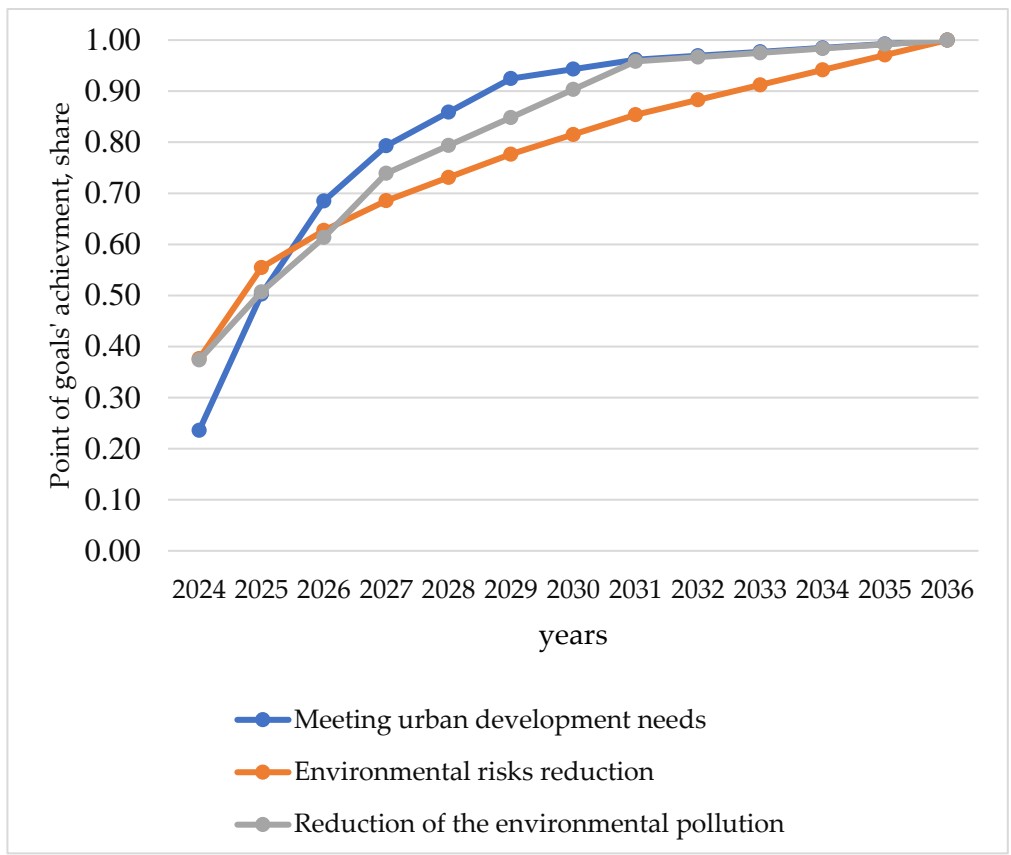

**Figure 5.** The achieved social development and environmental security goals by the mining dumps' development according to the obtained order.

An attempt to include the dump N2 in the 2024 plan with a priority of 0.15 is not possible since the development of this mountain dump requires USD 36 million. This sum exceeds the funding remains for 2024. The next priority for mining dump N4a use should be chosen. It requires a capital cost of USD 12 million and two years of development. This mining dump will be added to the 2024–2025 plan. Funding balances are being adjusted $B_{2024} = 16 - 12 = 4$: USD million; $B_{2025} = 27 - 12 = 15$ USD million. From 2025 to 2026, it is not possible to include the remaining dumps in the formed plan because the funding is insufficient. In 2027, the volume of possible financing is USD 27 million. Therefore, the mining dump N8a is included in this period (till 2031).

The adjustment to residual funding for this period is as follows: $B_{2027} = 27 - 25 = 2$ USD million; $B_t = 44 - 25 = 19$ USD million for $t = 2028, 2029$; and $B_t = 62 - 25 = 37$ USD million for $t = 2030, 2031$. The start of the N2 mining dump development is possible only from 2030 (Table 6).

**Table 6.** Order for mining dumps with the maximum volume of capital investments in USD 62 million for the mining maintenance and development.

| Analyzed Dumps | 2024 | 2025 | 2026 | 2027 | 2028 | 2029 | 2030 | 2031 | 2032 | 2033 | 2034 | 2035 | 2036 |
|---|---|---|---|---|---|---|---|---|---|---|---|---|---|
| Dump N2 |  |  |  |  |  |  | ▉ | ▉ | ▉ | ▉ | ▉ | ▉ | ▉ |
| Dump N4a | ▉ | ▉ |  |  |  |  |  |  |  |  |  |  |  |
| Dump N8 | ▉ | ▉ | ▉ | ▉ |  |  |  |  |  |  |  |  |  |
| Dump N9 | ▉ | ▉ | ▉ | ▉ | ▉ | ▉ |  |  |  |  |  |  |  |
| Dump N8a |  |  |  | ▉ | ▉ | ▉ | ▉ | ▉ | ▉ |  |  |  |  |
| Dump N12 |  | ▉ | ▉ | ▉ |  |  |  |  |  |  |  |  |  |
| Dump N2b | ▉ | ▉ |  |  |  |  |  |  |  |  |  |  |  |

As a result, a sequence was obtained for the technogenic deposits' involvement considering all mountain dumps development by 2036. It is possible to reduce the period of mining dumps development in four years. There could be recommendations to increase the funding for the dump's development for the 2026–2029 period. As Figure 4 shows, the capital costs and production volume correlate providing that obtained order of technogenic resources involvement into economic circulation has been realizing. This ensures high production profitability.

The following trajectories show the achieved social development and environmental security goals by the mining dumps development according obtained order (Figure 5).

The progressive process of goals achieving could be seen on the graph:

- in 2030, (the middle of the mining dumps development period) the risk of environmental emergencies has been reduced by 81.5%;
- urban development through reclaimed areas is 94.3% satisfied;
- environmental pollution reduced by 90.3%.

The results of the research were used for development strategy for Erdenet Mining Corporation SOE, Erdenet city and its social and environmental sustainability.

## 4. Discussion

Currently, there are two world trends in subsoil use and mining. Firstly, there is a depletion of natural resources and the reduction in the minerals' content in the extracted ore in many deposits, primarily non-ferrous metals, copper, and gold. For example, at the Erdenet mining enterprise in Mongolia, at the beginning of the deposit development in 1978, the copper content in the mined ore ranged from 0.685% to 0.833%. By 2013, this figure decreased to 0.530% and currently is 0.45%. According to forecasts, by 2040, the ore content could be 0.21%. Naturally, a decrease in the mineral content in the mined ore entails additional production and environmental costs for the enterprise. To maintain copper production volumes, this enterprise has to mine and to process more ore that is associated with an increased impact on the environment.

The deposit depletion can directly lead to the closure of a mining, the cessation of the operation of the enterprise itself as a whole, and be accompanied by adverse environmental and social consequences in the form of accumulated waste and loss of employment for many miners and their families. Note that the city of Erdenet (83.5 thousand inhabitants, 2023) in Mongolia is a single-industry town, which is highly dependent on the enterprise employing more than 6000 people.

Secondly, there is a steady tendency for the price of metals being based on global demand for copper and other non-ferrous metals, and it continues to increase. This has a positive impact on the activities of mining enterprises in the context of depletion of their resource base and stimulates their use of a new resource base—the so-called "substitute resources", i.e., resources of technogenic deposits with a lower content of minerals. They were formed as a result of past economic activities. For example, during the operation of the Erdenet mine, more than 28 million tons of ore were processed and the waste can largely be reused in production.

The scheme of the technogenic resources' involvement into economic turnover is shown in Figure 6. The calculations have shown that this process makes it possible to extend the operating time of the enterprise by 40 years, even when the mining concludes. Additionally, it ensures economically profitable operation of the enterprise by using secondary raw materials. This problem-solving approach considers many environmental problems, including liquidation, processing and use of mountain dumps, reclamation of disturbed lands, the use of cleared areas for housing construction and social infrastructure of the city, and preserving jobs.

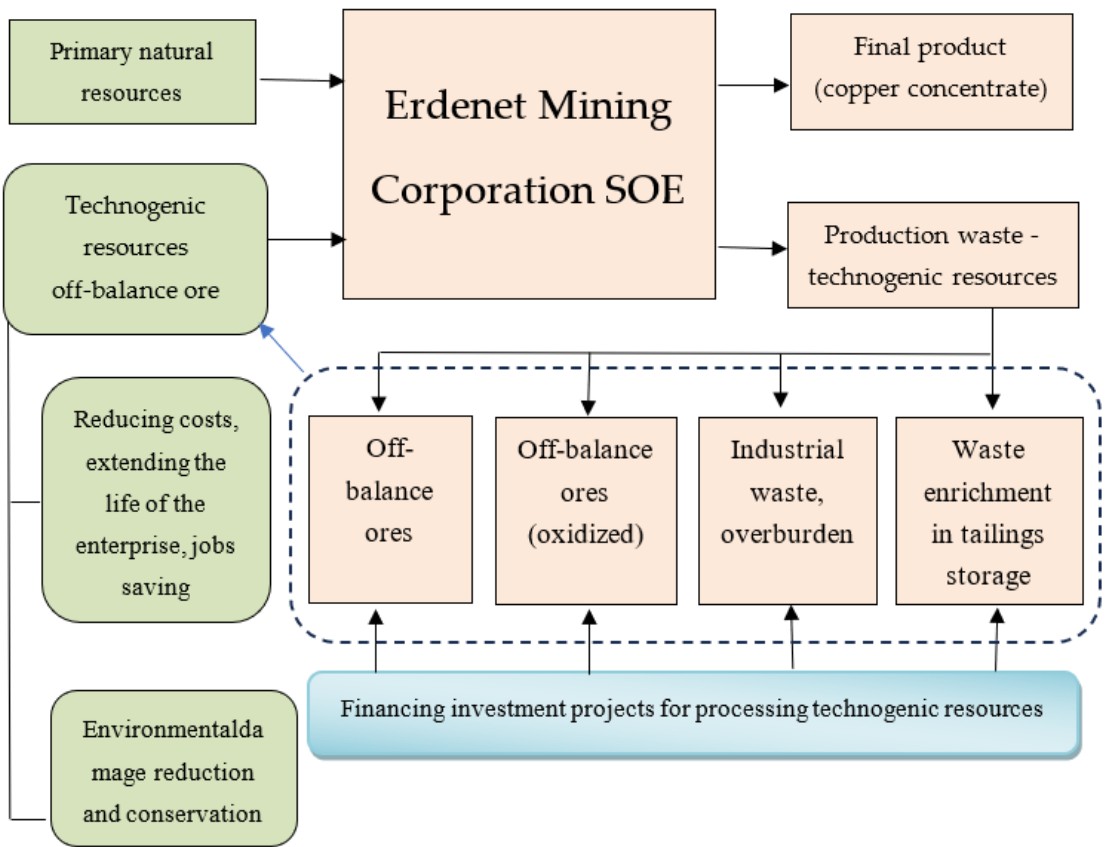

**Figure 6.** The scheme of the technogenic resources' involvement into economic turnover.

The proposed scientific tools and algorithms for selecting investment projects through including technogenic deposits in economic circulation can be used by various target groups of stakeholders using economic, environmental, social criteria.

So, for example, for a mining enterprise (for business purposes), this will enable the selection of the optimal development option, the selection of the stage the resources of technogenic deposits in order to ensure the necessary production profitability, growth of the resource base, and an increase in the life cycle of the enterprise.

At the same time, experts can use the proposed approaches to assess and analyze various alternative options for the development of a given territory from the standpoint of ensuring its sustainability.

For the city of Erdenet and its population, the proposed scientific tools make it possible to minimize environmental and social risks associated with the activities of this mining enterprise in the context of the depletion of its resource base. The proposed approach allows us to broadly predict economic, environmental, and social changes associated with the processing of mining waste to ensure future production diversification and sustainability.

## 5. Conclusions

The possible exhaustion and depletion of mineral reserves is among the factors affecting the functioning of mining enterprises. It could lead to the closure of a mine and the liquidation of an enterprise as a whole. This may cause significant environmental and social risks.

One of the directions for problem-solving is the use of resources from technogenic deposits. It could be formerly accumulated waste during ore mining and beneficiation, i.e., using previously accumulated waste with lower mineral content to produce useful products, such as copper. The case of Erdenet in Mongolia shows that accumulated technogenic waste can currently be considered as a resource base, a kind of "technogenic deposit". They could be involved into economic circulation, for example, for the production of useful products (copper, molybdenum), as well as be used as a secondary resource for the production of building materials, such as for road construction.

At the same time, for the socio-ecological assessment of the territory and the elimination of accumulated rock dumps, it is proposed to use criteria such as the impact on the environment, the demand for the territory for city development (for the construction of housing and social facilities), and the risks of emergency situations.

The analysis of the Erdenet mining enterprise for 2012–2022 showed a significant impact of world demand for copper and world prices for copper concentrate on the profitability of developing resources from technogenic deposits with lower mineral content. At the same time, the SWOT analysis results showed that the disadvantages of the activity of this mining enterprise include the depletion of its resource base, the implementation of compensatory measures if the mine is closed, as well as the increase in environmental costs.

The research proposes an algorithm and the model to determine the selection of the highest priority technogenic deposits, considering the volume of attracted investments, environmental, and social factors. The calculations have proved that using resources from technogenic deposits (waste), i.e., substitute resources with a low mineral content, in conditions of stable demand and high prices for copper on the world market, high production profitability is ensured.

The approach supposes the involvement of technogenic deposits as a new resource base for copper mining enterprises that makes it possible to extend its life considering the depletion of ore, to obtain useful products by recycling waste, and to reduce the environmental pollution. It also helps to decrease social risks through keeping local jobs and ensuring sustainable development. The developed approach has been justified upon the case of the Erdenet mining enterprise in Mongolia—one of the world largest copper mine developers. The result of the study is precisely the development of a model for the replacement of primary natural resources with secondary ones (resources of technogenic deposits). The developed approach was designed to define the priority order of dump mining development to achieve the desired economic, social, and environmental effects. The individual sets of criteria can be used for a specific production facility. Its composition should be dictated by geographical features, occupied dump areas, percentages of mined ore, the impact of dumps on the environment, and other factors. The research has proved that the technogenic deposits have a significant impact on the economics of mining enterprises and could be used for environmental improvement through the use of non-renewable natural resources. The proposed approach offers a mechanism for cost reduction by using technogenic raw materials for the production of finished products. With this mechanism, when the resource base is depleted, the mining company has an opportunity to prolong its activity by retaining higher profits and employees. The development of mining dumps for the production of building materials (sand, crushed stone, and gravel) could be the needed problem-solving solution through liquidation of accumulated environmental damage.

The results of the research on the use of substitute resources by using technogenic deposits instead of primary raw materials, which tend to be exhausted and depleted, are universal in nature and can be used at copper mining enterprises in the USA, Chile, Congo, Zambia, Botswana, and in other countries.

**Author Contributions:** Conceptualization, I.P. and A.N.; Validation, I.N., V.G.; Formal analysis, A.N. and I.N.; Resources D.N.; Data curation, V.G. and D.N.; Writing—original draft, I.P.; Writing—review & editing, V.G.; Visualization, A.N., V.G.; Supervision, I.P.; Project administration, V.G. All authors have read and agreed to the published version of the manuscript.

**Funding:** This research received no external funding.

**Data Availability Statement:** Data are contained within the article.

**Conflicts of Interest:** The authors declare no conflict of interest.

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
