# Peer review of "The Development of Technogenic Deposits as a Factor of Overcoming Resource Limitations and Ensuring Sustainability (Case of Erdenet Mining Corporation SOE in Mongolia)"

_sustainability, doi:10.3390/su152215807_

Round 1

Reviewer 1 Report

Comments and Suggestions for Authors

The utilization of mine waste for value-added mineral recovery has been a major research topic across the globe for a long time. It is not a new idea to use waste to recover additional materials. The authors also proved that with many in-text references to previously published studies. However, the outcomes of those studies weren't discussed in detail, and the feasibility of this idea is not clear.

In this reviewer's opinion, the introduction section is confusing the most. It doesn't emphasize the novelty of this manuscript, and it doesn't discuss the literature from an end-result point of view. No recovery results or potential process flowsheets are discussed. 

Moreover, the manuscript is too specific to the given mine. It sounds like a market analysis report for the Erdenet Mine.

A better-defining title could have been better (e.g., market analysis for copper dumps, etc.).

The language is quite good except for some minor issues. 

Please include axis titles and units for Figure 4.

Please define all the terms in the equations. 

Comments on the Quality of English Language

It is pretty good except for some minor grammar mistakes but acceptable. 

Author Response

Dear Reviewer!

Thank you for your attention to the manuscript and the remarks. We were doing the best to consider them during editing the paper for second round. The Introduction section has been updated as well as Abstract. Part of the literature that related to the use of waste in the extraction of gold and ferrous metals has been reduced. The statement of the problem and the novelty of the research are concreated and defined as to the mechanisms’ development of using technogenic deposits for copper production.

The purpose of the study, its novelty, is to substantiate the approach to the technogenic deposits involvement as a new resource base of a copper mining enterprise, which allows extending its life, to obtain useful products by recycling waste within the framework of a circular economy cycle, to reduce the burden on the environment, as well as to ensure sustainable development of the territory and save jobs. This approach is implemented using the case of the Erdenet mining enterprise in Mongolia.

The result of the study is precisely the development of a model for replacing primary natural resources with secondary ones (resources of technogenic deposits), as well as the substantiation of the mechanisms for implementing this approach using the case of the Erdenet enterprise. The article is devoted to the development of mechanisms and models for the use of man-made dumps, i.e., secondary resources that were formed as a result of the activities of the mining enterprise in previous years. The title of the article has been clarified taking into account the reviewer’s comment: The technogenic deposits’ development as a factor of overcoming resource limitations and ensuring sustainability (using the case of Erdenet Mining Corporation SOE in Mongolia).

Sincerely yours,

The Authors

Reviewer 2 Report

Comments and Suggestions for Authors

The manuscript needs substantial clarifications and improvement before being considered for publication.

1.       Section Introduction: (1) For easier understanding, please clarify the relationship between important terms such as waste, technogenic deposits, and secondary material resources, which are used in the first two paragraphs. (2) The literature review is too lengthy. Please abridge it by focusing literature on mining waste, especially copper ore mining waste, and clarify the research gap of the literature. (3) Please add one or two lines, at the end of the section, to clearly state the research aim of the manuscript.

2.       Section Materials and Methods: This section is quite difficult to follow. Please divide it into subsections and accordingly provide the title of each subsection. In addition, explanation of the AHP method should be shortened by simply referring to related literature.

3.       Section Results: This section is also not easy to follow. Please consider dividing it into subsections. Furthermore, in-depth discussion on the results is missing.

4.       Section Conclusion: Without explicitly stated research aim or in-depth discussion of research results, the conclusion is rather unconvincing.

Comments on the Quality of English Language

The English language is basically OK to the reviewer. 

Author Response

Dear Reviewer!

Thank you for your attention to the manuscript and the remarks. We were doing the best to consider them during editing the paper for second round. The Introduction has been updated. The purpose and novelty of the research were clarified. In this article there is the following relationship between the concepts. As a result of ore mining and processing, useful products are obtained (copper concentrate) and waste is generated (overburden waste and enrichment waste). In turn, waste from overburden rocks at the time of ore mining in previous years contained a low content of minerals (copper and molybdenum). Now, due to the advent of new technologies for the extraction of copper and molybdenum, the development of such waste is becoming profitable. In other words, previously accumulated rock dumps are considered in the article as new technogenic deposits, i.e. as new resources that were formed on the basis of previously accumulated waste. Thus, the secondary resources in the article are also the resources of such technogenic deposits, which are used both to obtain useful products (copper) from raw materials with a lower mineral content, and to use such waste for the production of building materials, for the construction of roads, etc. What is new in the article is the replacement of primary natural resources that run out at a given deposit, the ore mine is closed, and the use of resources of technogenic deposits, i.e. former waste, which acts as new raw material for obtaining useful products (copper and molybdenum). The possibility of using the resources of mineral deposits with a relatively low mineral content is associated with rising prices for copper on the London Commodity Exchange, with increasing demand for copper in the world, which makes it possible to economically process former waste. At the same time, environmental and social problems are solved.

In the Introduction section, the literature is clarified. Sources that are not related to copper mining have been reduced, for example, sources related to gold and iron ore deposits. The problem in the scientific literature is connected precisely with the study of the possibility of using the resources of technogenic deposits as a substitute for primary natural raw materials. The primary raw material, ore at the deposit, is depleted and it becomes profitable to use new technogenic resources that were formed through the accumulation of former waste with a low mineral content. This issue has not been sufficiently studied in the scientific literature, especially in the context of the connection between the use of waste, reducing the environmental pollution, increasing demand and prices for copper on world markets, and extending the life of the enterprise, even if the ore mine is closed. The proposed approach, which was implemented at the Erdenet enterprise in Mongolia, is important for sustainability: economic, environmental and social. This is the importance and scientific significance of the work. In the Introduction section, the purpose of the manuscript is formulated: The purpose of the study is to substantiate the approach for technogenic deposits involvement as a new resource base of a copper mining enterprise, which allows its life ex-tending, useful products’ processing through waste management and circular economy development, reducing of the environmental pollution, as well as to ensure sustainable development and to keep local jobs. The proposed approach is justified upon the case of the Erdenet mining enterprise in Mongolia. Comments regarding the mathematical part of the study have been fully taken into account.

Reviewer 3 Report

Comments and Suggestions for Authors

1. The paper focuses on copper production, and the title of the paper does not substantially reflect the content of the paper.

2. The rearrangement of abstract and conclusion sections are necessary. In abstract section, it is improper to describe the experiment process using the structure of “The authors analyze”, “The authors use”, etc. The conclusion section is short and it is suggested that the content in conclusion section should be divided into two to three paragraphs. The interesting and important results should be placed in the abstract and conclusion sections. References should be cited in the conclusion section.

3. There are many short paragraphs in the introduction section, some should be merged. In the Introduction section, although a detailed review on waste accumulation was provided, some references on mining waste processing should be cited, e.g., DOI: 10.1016/j.mineng.2023.108296; 10.1016/j.fuel.2022.126988.

4. Figs. 1-6 are unattractive and should be redrawn. There is no scale on the horizontal coordinate.

5. The format of the paper should be consistent.

6. There are some syntax errors, sentence errors and other English Language errors that must be corrected. Possibly, a native English Language speaking Scientists should be employed for the final editing.

Comments on the Quality of English Language

There are some syntax errors, sentence errors and other English Language errors that must be corrected. Possibly, a native English Language speaking Scientists should be employed for the final editing.

Author Response

Dear Reviewer!

Thank you for your attention to the manuscript and the remarks. We were doing the best to consider them during editing the paper for second round.

The title of the article has been adjusted taking into account the reviewer's recommendations. The technogenic deposits’ development as a factor of overcoming resource limitations and ensuring sustainability (case of Erdenet Mining Corporation SOE in Mongolia). The comments regarding the mathematical part of the study have been fully taken into account. The recommended references were attentively researched and included to the reference review and list. The Introduction has been updated. The purpose and novelty of the research were clarified. In this article there is the following relationship between the concepts. As a result of ore mining and processing, useful products are obtained (copper concentrate) and waste is generated (overburden waste and enrichment waste). In turn, waste from overburden rocks at the time of ore mining in previous years contained a low content of minerals (copper and molybdenum). Now, due to the advent of new technologies for the extraction of copper and molybdenum, the development of such waste is becoming profitable. In other words, previously accumulated rock dumps are considered in the article as new technogenic deposits, i.e. as new resources that were formed on the basis of previously accumulated waste. Thus, the secondary resources in the article are also the resources of such technogenic deposits, which are used both to obtain useful products (copper) from raw materials with a lower mineral content, and to use such waste for the production of building materials, for the construction of roads, etc. What is new in the article is the replacement of primary natural resources that run out at a given deposit, the ore mine is closed, and the use of resources of technogenic deposits, i.e. former waste, which acts as new raw material for obtaining useful products (copper and molybdenum). The possibility of using the resources of mineral deposits with a relatively low mineral content is associated with rising prices for copper on the London Commodity Exchange, with increasing demand for copper in the world, which makes it possible to economically process former waste. At the same time, environmental and social problems are solved.

In the Introduction section, the literature is clarified. Sources that are not related to copper mining have been reduced, for example, sources related to gold and iron ore deposits. The problem in the scientific literature is connected precisely with the study of the possibility of using the resources of technogenic deposits as a substitute for primary natural raw materials. The primary raw material, ore at the deposit, is depleted and it becomes profitable to use new technogenic resources that were formed through the accumulation of former waste with a low mineral content. This issue has not been sufficiently studied in the scientific literature, especially in the context of the connection between the use of waste, reducing the environmental pollution, increasing demand and prices for copper on world markets, and extending the life of the enterprise, even if the ore mine is closed. The proposed approach, which was implemented at the Erdenet enterprise in Mongolia, is important for sustainability: economic, environmental and social. This is the importance and scientific significance of the work. In the Introduction section, the purpose of the manuscript is formulated: The purpose of the study is to substantiate the approach for technogenic deposits involvement as a new resource base of a copper mining enterprise, which allows its life ex-tending, useful products’ processing through waste management and circular economy development, reducing of the environmental pollution, as well as to ensure sustainable development and to keep local jobs. The proposed approach is justified upon the case of the Erdenet mining enterprise in Mongolia. Comments regarding the mathematical part of the study have been fully taken into account.

Sincerely yours,

The Authors

Round 2

Reviewer 1 Report

Comments and Suggestions for Authors

Clearly, the authors spent a lot of effort addressing the comments and suggestions. However, there is one issue that needs urgent correction. On page 11, the first few sentences are in Russian. Please correct that.  

Comments on the Quality of English Language

Except for that one issue mentioned above, the English language is good. 

Author Response

Dear Reviewer!

We are sincerely grateful for your high qualified attention to our manuscript. The recommendations you’ve given made the paper more sophisticated and deeper. We are glad that it has been succeed to keep the balance on critical reviewing and evaluating of scientific novelty. 

Sincerely yours,

The Authors.

Reviewer 2 Report

Comments and Suggestions for Authors

1. It is hard for the reviewer to tell how exactly the authors have revised the manuscript. The authors should provide a point-by-point reponse to the reviewer's previous comments. 

2.  In-depth discussion on the results is still missing. Please re-organise the content of section 3 and provide the discussion in a seperate section. 

Author Response

Dear Reviewer!

We are sincerely grateful for your high qualified attention to our manuscript. The recommendations you’ve given made the paper more sophisticated and deeper. We are glad that it has been succeed to keep the balance on critical reviewing and evaluating of scientific novelty. 

We were following all your remarks:

  1. The Discussion part has been added to the manuscript to develop in-depth discussion. We tried to re-organized the content of section 3 to provide the discussion in a separate section (pages 15-17) 
  2. In the Introduction section, the literature is clarified. Sources that are not related to copper mining have been reduced, for example, sources related to gold and iron ore deposits. The problem in the scientific literature is connected precisely with the study of the possibility of using the resources of technogenic deposits as a substitute for primary natural raw materials. (pages 1-4)
  3. The scientific novelty was rewritten. What is new in the article is the replacement of primary natural resources that run out at a given deposit, the ore mine is closed, and the use of resources of technogenic deposits, i.e. former waste, which acts as new raw material for obtaining useful products (copper and molybdenum). The possibility of using the resources of mineral deposits with a relatively low mineral content is associated with rising prices for copper on the London Commodity Exchange, with increasing demand for copper in the world, which makes it possible to economically process former waste. At the same time, environmental and social problems are solved.
  4. Comments regarding the mathematical part of the study have been fully taken into account.

Hope we have answered to your comments in the last edited version of the manuscript.

Sincerely yours,

The Authors.

Round 3

Reviewer 2 Report

Comments and Suggestions for Authors

A more point-by-point reply has been provided and the substantial comment on an in-depth discussion has been added. The reviewer would suggest publication of the revision. 

Comments on the Quality of English Language

further English-language proofread is suggested. 

Author Response

Dear Reviewer! We thank you for your attention to our manuscript. Your recommendations made it possible to make it better and more scientific. You were absolutely right in advising us to add a Discussion section. This allowed for a more in-depth analysis of possible uses of the proposed model. Now the article looks more structured. Improvement of English is expected during final editing of the text before publication. Thank you again for your excellent advice and comments.